# Local Regularisation in Histological Registration for Preserving Cell Morphology

**Anonymous Author**                                              ANON@ANON.AC.UK
*Department of Secrecy*
*University of Nowhere*
*Area 51*

**Editor:**

## Abstract

Histopathological diagnoses may depend on both the presence/absence of specific biomarkers, as well as cell morphology. When registering histology images, due to structural differences in consecutive slices or damage during staining, affine transforms may fail to align high-level structures. As such, non-rigid registration can be used to align such structures. This can distort cell morphology, which may be problematic when a diagnosis or inference depends on such. We introduce a regularisation approach in which a cell segmentation induces a *cellularity* image, indicating a regions degree of cellularity. This is used to weight a pixel-wise regularity term which encourages such regions to be rigid, while permitting non-rigid deformations elsewhere in the image. We show that under certain configurations this method results in no loss of accuracy (as measured by annotated keypoint distances), with potential improvements to measured levels of non-rigidity in cell regions.

**Keywords:** Whole-Slide Image Registration, Non-Rigid Registration

## 1. Introduction

Histopathological diagnoses may rely on information made available in multiple staining and/or imaging media. For instance, a tumour diagnosis may rely on both the expression of a certain antigen (e.g. PSA), along with cell morphology or other local structural properties - in a specific tissue region. In order to consider this plurality of information sources within a specific spatial region (i.e. the site of a candidate tumour) across several images, the pathologist must identify which regions correspond in each image. While this can be done manually by eye, it may also be done computationally using Image Registration, and indeed further diagnostic medical tasks may be automatically performed on aligned image sets.

General purpose registration algorithms constrain the transformation which is found in many different ways, though usually this takes the form of a global constraint on something analogous to a material property, e.g. elasticity. In doing so, large-scale deformation may be controlled, which in some instances may be desired, though distortion at a smaller scale may not be considered. Given morphological information can be key to diagnosis, we argue that in a histopathological setting of registration, regularisation should be considered with cellular - rather than tissue-level - granularity. We propose the use of a regularisation penalty during registration which penalises deformations specifically over (automatically detected) cellular regions, in order to allow large-scale alignment of structures while preserving cellular

morphological integrity. We show configurations of this procedure in which distortion to cellular regions is reduced, at no statistically significant cost to the overall registration accuracy (as measured by final distance between annotated keypoints after registration). The ANHIR Borovec et al. (2020) dataset is used to evaluate our method.

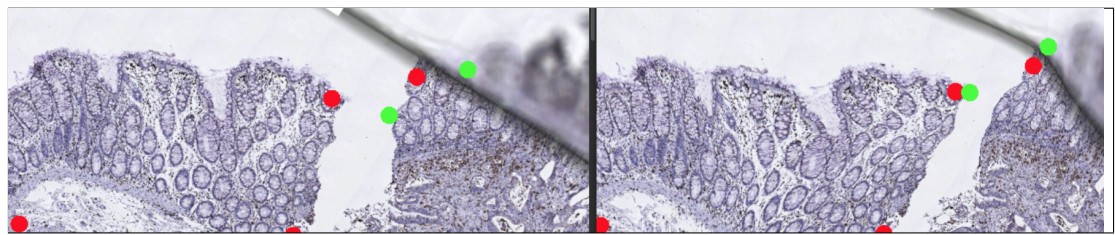

Figure 1: An example of distortion to cell morphology. Left shows tissue under the optimal affine registration, whereas right side shows the same tissue after non-regularised non-rigid registration. The green and red circles indicate hand-labelled corresponding anatomical regions. We see that while these keypoints align more closely after non-rigid registration, the shapes and sizes of cells in the vicinity have been altered. Our approach seeks to not compromise keypoint accuracy while reducing cell distortion.

## 2. Related Work

**Variational Image Registration**  The family of image registration approaches, known as 'intensity-based', 'optimization-based', or 'variational' approaches are unified and given thorough exposition in Fischerand and Modersitzki (2003), herein we refer to this formulation of registration as Variational Image Registration. Within this framework for registration the problem is characterised by the minimization of a functional[1] made up of an error term (e.g. SSD) and a regularization term, which penalises transformations contravening desired properties. These functions may be physically inspired, imagining the image we wish to transform being somehow 'attached' to a physical entity and constrained to behave as such. E.g. Elastic regularization, as in Fischler and Elschlager (1973); Broit (1981). The elastic regularizer uses the mathematical characterisation of strain in an elastic material to penalise deformations which 'squeeze', 'twist', and 'distort' too greatly.

Many physically inspired regularizers have been explored, many of which are formulated in terms of multi-variate derivatives (e.g. curl, divergence, gradient) Fischer and Modersitzki (2004). Notable examples of such physically inspired regularizers include bending-energy where we can imagine the image on a semi-rigid sheet of metal Bookstein (1989); fluid-based where we can imagine the image as viscous paint Christensen (1994); and diffusion where we can imagine the image as a coloured gas Fischer and Modersitzki (2002).

**Image Registration in a Histological Context**  For images with are similar in appearance, optimisation using the SSD for registration is perfectly viable - intuitively perfect alignment results in all image intensities matching. However, in a histological setting variation in appearance is particularly common - agreement of image intensities does not imply

---

1. A function whose argument is itself, a function. Here the transformation function which warps between images to be registered.

good registration. One solution for this is use of a cost function which is invariant to changes in intensity. To this end Haber and Modersitzki (2006) introduced the Normalised Gradient Field loss, originally for the multi-modal registration of MRI, CT, and PET images. The loss has a particularly simple interpretation, the contribution at each spatial location is given by the cosine distance (angular disagreement) of the image derivatives at that point between the two images, i.e. the loss penalises images whose intensities vary in different directions. This loss was used for registration of histological images twice in 2019 by Bulten et al. (2019); Lotz et al. (2019). This type of approach is currently SOTA on the ANHIR dataset Borovec et al. (2020).

The notable problems in histological data of variation in appearance, artefacts, and non-rigid deformations are recognised by the authors in Wang and Chen (2013) who develop a robust alignment procedure for H&E stained histological sections. They place a particular emphasis on colour-normalization, intending to reveal salient structural tissue patterns. They use colour-deconvolution to select colours which they posit to correspond to 'eosinophilic' structures, i.e. tissue, but not nuclei. They claim isolating these structures benefits feature-matching. They extract interest points and pair them using a DoG operator followed by RANSAC Fischler and Bolles (1981), and minimize an error function containing terms pertaining to image intensities, intra-pair distances, and geometric consistency. In a second stage they minimize SSD with elastic regularization w.r.t the parameters of a deformation field, initialised using the pair-correspondences estimated in the first stage.

Clearly colour and spatial variation are a significant issue in histological registration. The approach of Kuska et al. (2006) attempts to sidestep this by performing registration guided not by differences in image intensity, but disagreement in segmentations. By doing so they align higher-level structures rather than promoting agreement on a pixel-level. Features are extracted densely, and clustered using a Gaussian mixture model. Following this, they follow the variational method of Amit (1994) and optimise the parameters of a displacement field such that the two segmentation images agree and a smoothness conditional is met. Other authors have met with success using segmentation to guide registration, with similar methods proposed in Borovec et al. (2013); Kybic and Borovec (2014).

There are scenarios which do not use segmentation for registration directly, but which perform segmentation in order to remove the foreground object and then follow an optimisation-based approach with the optimisation criterion only defined over the foreground region, as in Dauguet et al. (2007); Shojaii and Martel (2016). Whether segmentation *per se* is used for registration, or as a smaller piece of a larger pipeline it does appear to be an effective way to remove the complications associated with background noise and artefacts which may be present in histological images.

## 3. Method

Like some previous authors, segmentation plays a key role in our registration pipeline. We however utilise it not for alignment, but for regularisation. First obtaining cell masks through segmentation (binary images indicating the presence of cells), and subsequently post-processing these masks to form *cellularity images*, real-valued images whose value indicates the *degree* of cellularity across a wider *region* not just as the cell location.

### 3.1 Cell Masks

In order to obtain the raw cell masks for all H&E stained images, we use the inbuilt method within QuPath, an open-source image analysis package for digital pathology Bankhead et al. (2017). This method is a variant on the Watershed algorithm for segmentation Digabel and Lantuéjoul (1978), though our method is agnostic to the sources of the original masks.

We attempt three different types of preprocessing on each mask, producing a set of (no-longer binary) 'mask' images, which capture the degree of cellularity, as opposed to a pixel-wise binary indicator. Firstly we use a Gaussian blur of the mask image, which preserves modes at the cell locations while indicating that nearby pixels also belong to cellular regions, the degree to which cell locations are directly emphasised, and the size of the spread of which nearby regions are considered close enough to be 'cellular' is governed by the filter's kernel size. Next, we simply perform a box blur, convolving with a kernel of all 1's (as the mask images are normalised, the scaling factor on the kernel is irrelevant). This produces an image where each pixel is replaced by a *count* of cellular pixels in its local vicinity (determined by the kernel size). Lastly, we employ the euclidean distance transform (EDT) to the raw cell masks, replacing each pixel with its euclidean distance from the closest pixel belonging directly to a cell. We term these images *cellularity images* and use each one as a pixelwise weighting on our regularity term during registration, the conjecture being that weighting cellular regions highly (i.e. penalising non-rigid deformations at those locations more), we will obtain a registration which better preserves morphology over cellular vs. non-cellular (or *lesser-cellular*) regions. We refer to the basic binary mask as the *raw* or *rigidity* mask; the euclidean distance transform as the *distance* mask; the gaussian blurred mask as the *gaussian* mask; and the box blurred mask as the *region count* or *box* mask.

### 3.2 Registration

Before performing the non-rigid registration, the optimal pre-registration is found (in closed-form). We consider the optimal pre-registration to be the similarity transform $T \in Sim(2)$ which minimizes the squared distance between pairs of corresponding labelled keypoints.

We employ non-rigid registration using the framework of 'variational image registration', with our transformation function induced using b-splines. For this, we use the elastix image registration toolkit Klein et al. (2010). For our rigidity penalty we use that of Starting et. al Staring et al. (2006), which enforces local-rigidity using three weighted constraints: affinity, orthonormality, and properness, discussed in subsection 3.4.

### 3.3 Data Term

Let $(I_F(\mathbf{p}), I_M(\mathbf{p}))$ denote a pair of mappings $I : \mathcal{P} \to \mathbb{R}$ representing images as functions mapping a spatial location to an intensity, where $\mathbf{p} \in \mathcal{P}$ indicates a spatial index over a space $\mathcal{P}$, for example $\mathbb{R}^2$ for 2D images. Let $d(I_F(\mathbf{p}), I_M(\mathbf{p}'))$ be a dissimilarity function mapping a pair of intensities to a dissimilarity score. A simple example of dissimilarity function is squared difference $d(x, y) = (x - y)^2$. Image registration may be phrased mathematically as an optimisation problem, where we adjust the parameters $\theta \in \Theta$ of a spatial transformation $T : \mathcal{P} \times \Theta \to \mathcal{P}$ to minimize an aggregated dissimilarity (loss function) over the fixed image

and the transformed moving image:

$$L(\theta) = \sum_{\mathbf{p} \in \Omega} d(\mathrm{I}_F(\mathbf{p}), \ \mathrm{I}_M(T(\mathbf{p}, \theta))). \tag{1}$$

Here $\Omega = \mathrm{Image}(\mathrm{Dom}(\mathrm{I}_F), T(\cdot, \theta)) \cap \mathrm{Dom}(\mathrm{I}_M)$ is the region where the two images overlap after warping, ergo the 'image' of the fixed image's domain under the transform intersected with the moving images domain. Minimizing $L(\theta)$ corresponds to the warped moving image's intensity agreeing with that of the fixed image. It is worth noting that as we typically have images discretised on a grid that a transformed point may not land on a grid location and so its value must be obtained by interpolation.

This 'data-term' is added to a regularity penalty $R$, weighted by a parameter $\gamma$,

$$J(\theta) = L(\theta) + \gamma R(\theta), \tag{2}$$

in our setting the overall strength of the regularity penalty is controlled by $\gamma$, but also influenced through use of the *cellularity image* which further weights this penalty *pixel-wise* based on the degree of 'cellularity' at the pixel in question.

We characterise 'error' using the mTRE criterion, which is a measure of the proximity of pairs of annotated keypoints, given by

$$\mathrm{mTRE}_{ij} = \frac{1}{\Psi_i} \mathrm{median}\Big( \{\|\mathbf{p} - \mathbf{q}\|_2 \mid (\mathbf{p}, \mathbf{q}) \in \Phi_{ij}\}\Big), \tag{3}$$

where $\mathbf{p}$ and $\mathbf{q}$ indicate keypoints of the fixed and moving image respectively, $\Phi_{ij}$ is the set of keypoint pairs for images $i$ and $j$, and $\Psi_i$ is the diagonal length of image $i$.

### 3.4 Rigidity Penalty

Our transformation maps pixels from a fixed image $I_F(\mathbf{p})$ into a moving image $I_M(\mathbf{p}\prime)$ like so $T : \mathbf{p} \mapsto \mathbf{u}(\mathbf{p}) + \mathbf{p}$. Where $\mathbf{u}(\mathbf{p})$ is given by an induced deformation field, to be represented by a lower-dimensional field upscaled with b-splines. The penalty includes several conditions upon $T$ for rigidity in a neighbourhood around a point $\mathbf{p}$.

**Affinity** If the transformation is to be locally rigid it must represent a transformation in $\mathbb{SE}(2)$. A necessary, but not sufficient condition for this is for $T$ to be affine, i.e. $T : \mathbf{p} \mapsto \mathbf{R}\mathbf{p} + \mathbf{t}$.

If $T$ takes this form then Hessian matrix of second order derivatives of $\mathbf{u}(\mathbf{p})$ should be the zero matrix. $\mathbb{H}[\mathbf{u}(\mathbf{p})] = \mathbf{0}$. Per coordinate they use $\left(\frac{\partial \mathbf{u}_k(\mathbf{p})}{\partial \mathbf{p}_i \partial \mathbf{p}_j}\right)^2$, $\forall i, j, k \in \{0, 1, 2\}$ giving the penalty then as the squared deviation of each hessian component from 0.

**Orthonormality** This penalty is not sufficient to ensure that $T \in \mathbb{SE}(2)$. It must also be the case that the matrix $\mathbf{R} \in \mathbb{SO}(2)$, i.e. that it be orthogonal (have orthonormal columns). When this is the case we have $\mathbf{R}\mathbf{R} = \mathbf{I}$. Elementwise we have $\sum_{k \in \{1,2\}} \mathbf{R}_{ki} \mathbf{R}_{kj} = \delta_{ij}$ as the orthonormality condition.

Differentiating coordinate $i$ of $\mathbf{u}(\mathbf{p}) + \mathbf{p} = \mathbf{R}\mathbf{p} + \mathbf{t}$ with respect to $\mathbf{p}_j$ we obtain $\frac{\partial \mathbf{u}_i(\mathbf{p})}{\partial \mathbf{p}_j} + \delta_{ij} = \mathbf{R}_{ij}$ as the kronecker delta on the right hand side simply picks elements out of $\mathbf{R}$ this implies also, that $\mathbf{R} = \mathbb{J}[T]$, the jacobian of the transformation $T$. From this we

then have $\frac{\partial \mathbf{u}_i(\mathbf{p})}{\partial \mathbf{p}_j} = \mathbf{R}_{ij} - \delta_{ij}$. Substituting this into the orthonormality condition we have $\sum_{k \in \{0,1\}} \left( \frac{\partial \mathbf{u}_k(\mathbf{p})}{\partial \mathbf{p}_i} + \delta_{ki} \right) \left( \frac{\partial \mathbf{u}_k(\mathbf{p})}{\partial \mathbf{p}_j} + \delta_{kj} \right) - \delta_{ij} = 0, \forall i, j \in \{0, 1\}$. The square of each coordinate is added to overall penalty as the contribution for orthonormality.

**Properness** This scenario still permits the transformation to represent reflections. In order to prevent this, a final condition of $\det(\mathbf{R}) - 1 = 0$ is used to encourage the determinant of $\mathbf{R}$ (the jacobian determinant of $T$) to be positive and so only represent a rotation. This also encodes "incompressibility", i.e. that the transformation cannot change the volume around a point. This is not sufficient in itself, as it permits e.g. compressing by a half in one dimension and expanding by double in the other.

## 4. Experiments and Results

We perform registration between pairs of corresponding (but differently stained) images, each time using the non-rigid method described in section 3. We repeat this for each type of cellularity image, at four different levels of regularisation strength $\gamma$. We perform statistical tests for significant changes to accuracy (mTRE), along with measures of deformation.

### 4.1 Effect on Keypoint Accuracy

Our results show that for our lowest two choices of the regularisation parameter $\gamma$, (i.e. $10^{-4}$, $10^{-3}$) there is no statistically significant drop in keypoint accuracy (mTRE) when employing the regulariser, vs. the no-regularization baseline method for each choice of mask. Statistical significance is established through use of a bootstrap hypothesis test Davison and Hinkley (1997); Efron and Tibshirani (1993), this non-parametric test was chosen due to the non-normality of the distribution of keypoint errors (mTRE) as verfied throught the use of the Shapiro-Wilk test for normality SHAPIRO and WILK (1965). For all statistical tests the standard $\alpha = 0.05$ significance level was used-shown in table 1.

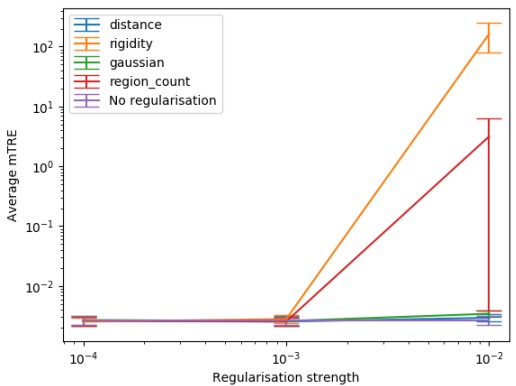

Figure 2: Keypoint accuracy for different mask types and values of $\gamma$.

### 4.2 Effect on Deformation Measures

In order to measure the deformation after registration for the different configurations of the cellularity images and regularisation strength we use two different pixel-wise deformation metrics. The first metric which we use is the determinant of the jacobian of the transformation as a given point. This tells us the degree to which a transform expands or compresses space around that point. The second metric we employ is the norm of the gradient of that quantity, which tells us the degree to which the expansion or compression factor is changing about a point. This essentially permits scaling to cellular regions, providing this scaling is

| Mask Type | $\gamma$ | $N$ | Critical Region | $\Delta$ mTRE | Significant |
|---|---|---|---|---|---|
| distance | $10^{-4}$ | 21 | (-1.59E-04, 5.65E-05) | -5.90E-05 | No |
| - | $10^{-3}$ | 21 | (-2.99E-04, 7.32E-05) | -1.18E-04 | No |
| - | $10^{-2}$ | 21 | (8.13E-05, 5.08E-04) | 2.84E-04 | **Yes** |
| - | $10^{-1}$ | 1 | (5.29E-04, 5.29E-04) | 5.29E-04 | **Yes** |
| raw | $10^{-4}$ | 21 | (-2.48E-04, 1.42E-05) | -1.29E-04 | No |
| - | $10^{-3}$ | 21 | (-4.94E-05, 3.40E-04) | 1.39E-04 | No |
| - | $10^{-2}$ | 19 | (-1.45E+01, 3.02E+02) | 1.64E+02 | No |
| - | $10^{-1}$ | 10 | (-3.74E+04, 1.72E+05) | 8.60E+04 | No |
| gaussian | $10^{-4}$ | 21 | (-1.96E-04, 2.34E-04) | 3.71E-05 | No |
| - | $10^{-3}$ | 21 | (-2.03E-04, 9.34E-05) | -6.62E-05 | No |
| - | $10^{-2}$ | 19 | (3.45E-04, 9.31E-04) | 6.28E-04 | **Yes** |
| - | $10^{-1}$ | 4 | (-9.59E+13, 1.92E+14) | 9.59E+13 | No |
| box | $10^{-4}$ | 21 | (-1.03E-04, 4.91E-05) | -3.10E-05 | No |
| - | $10^{-3}$ | 21 | (-2.69E-04, 6.97E-05) | -1.10E-04 | No |
| - | $10^{-2}$ | 21 | (-3.16E+00, 6.31E+00) | 3.16E+00 | No |
| - | $10^{-1}$ | 9 | (3.50E+13, 3.34E+14) | 1.95E+14 | **Yes** |

Table 1: When $\gamma \leq 10^{-2}$ we see no significant change to accuracy (mTRE) for any of the mask types. $N$ = number of paired samples which could be used for the bootstrap estimate. For higher values of $\gamma$ registration may fail, leading to fewer than all 21 pairs being available.

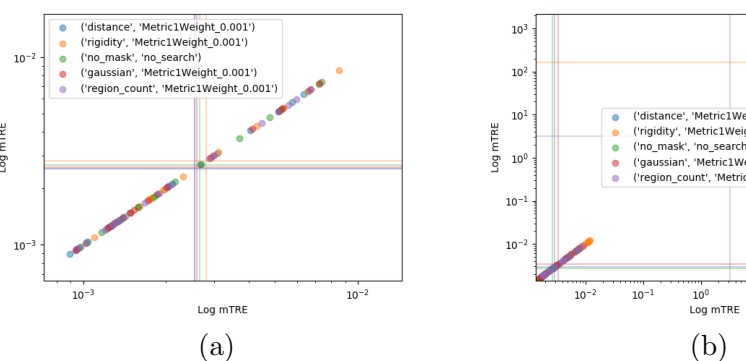

(a)  (b)

Figure 3: For sufficiently high values of $\gamma$ registration may fail, skewing summary statistics, as shown. Here we see the log mTRE scattered against itself, with vertical and horizontal lines showing the mean for that mask type. (a) shows this for $\gamma = 10^{-3}$, where log mTRE has no notable extreme values (given all registrations terminate properly). (b) shows that certain mask types lead to a small number of outliers which greatly distort the means, these are mask types *rigidity* and *region count*.

locally constant. As these two metrics are pixel-wise, we choose two schemes for aggregating them. Firstly, we sum these values over the location of cells. Second we sum them over the entire tissue image, this is found via the convex-hull of the annotated keypoints.

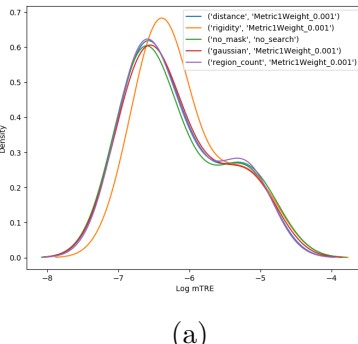 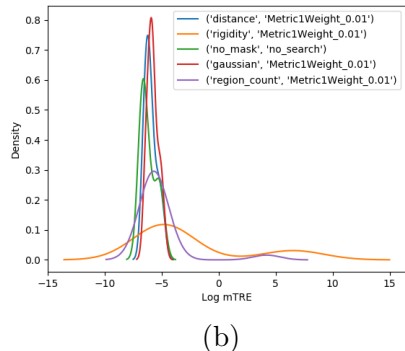

(a)          (b)

Figure 4: An alternative way to visualise the data in figure 3 is to fit kernel density estimators Rosenblatt (1956); Parzen (1962) to the (log) mTRE. This illustrates the degree to which a small number of extreme values distort the error distribution.

**Significant decreases in deformation** We find when $\gamma$ is sufficiently low, there are several configurations under which a statistically significant decrease in deformation is present. These are the *box* mask type, with $\gamma = 10^{-4}$, as measured by the jacobian determinant aggregated over the whole tissue region (Reduction of $-8.64 \times 10^{8}$, critical region $(-1.72 \times 10^{9}, -7.41 \times 10^{7})$); the *box* mask type, with $\gamma = 10^{-2}$, as measured by the norm of the gradient of the jacobian determinant aggregated over the whole tissue region (Reduction of $-5.45 \times 10^{6}$, critical region $(-1.02 \times 10^{7}, -8.74 \times 10^{5})$); and the *box* mask type, with $\gamma = 10^{-2}$, as measured by the norm of the gradient of the jacobian determinant aggregated over the cell masks (Reduction of $-1.46 \times 10^{8}$, critical region $(-2.62 \times 10^{8}, -4.64 \times 10^{7})$).

Otherwise, in no circumstances when $\gamma <= 10^{-2}$ do we see significant increase to any of the local deformation measures. Full tabular results with statistical tests as in table 1 will be provided in the appendix, but are omitted here for brevity.

## 5. Conclusion and Further Work

**Conclusion** Our work shows that for an appropriate selection of parameters our cell-local regularisation scheme using *cellularity images* may reduce deformation in cellular regions of histology images (and consequently a preservation of morphology in such regions) without a statistically signficant drop in overall registration accuracy, as measure by mTRE. As such, our method may be appropriate when performing registration of histological images with downstream tasks which rely on morphological properties of cells.

**Further Work** The ANHIR Borovec et al. (2020) dataset used in this paper features images which are not of sufficient resolution to measure morphological cell properties reliably (e.g. circularity, eccentricity), further work will involve evaluation of this and similar methods for local-regularisation based on region cellularity, but evaluated not only by measuring local deformation within regions using the registrations induced transformation, but measuring the changes in morphological metrics for cells before and after registration. As such we hope to support the hypothesis that such a regularisation strategy facilitates diagnosis and analysis for image pairs where morphological measurements are important.

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
