# OpenReview forum: "Local Regularisation in Histological Registration forPreserving Cell Morphology"
_MICCAI.org/2021/Workshop/COMPAY — Reject_

### Official Review · Reviewer_u3qq · 2021-08-17
**Very good idea / important problem but missing support**

**Rating:** 5
**Confidence:** 4

**Review:**

The authors deal with the important problem of pathology image registration, particularly how to preserve morphological structure more intelligently to not destroy diagnostically relevant information. This is a very interesting problem and also they show potentially an interesting approach.

Unfortunately, the dataset description (section 4) is incomplete. Which images did you register? Which staining? Coarse? Dense? How many image? How large? Further, the results do not support the main claim of the paper. Section 4.2. describes the metrics they want to measure to quantify the preserved cellular structures, but these dimensionless numbers are not descriptive. What do they mean? How do they compare to other methods (w/o regularization)? What is the impact of −8.64 × 10^8 reduction of Jacobian Determinant? Can you show some example images?

Still, I find their work important and also innovative, so worth to be published. But the current draft needs major revision to fix the missing parts.

Further concerns following:

Major issues:
- Please provide an overview of your method (e.g. figure). e.g. it seems in Section 3.2 that manual keypoints have to be set.
- Figure 1, 2, 3, 4 are not referenced
- Section 3.1: Please provide example images of the different masks you use. This is helpful for the reader to understand what the method is doing.
- Section 3.2: What is pre-registration doing? Just rigid transformation (translation, rotation...)? Do keypoints have to be manually set?
- Section 3.4. does not seem to be main contribution, so it could go to supplement to save space.
- Experiments section 4 is missing (Page 6). Dataset used? Example images?
- Figures need larger axes labels and legends. They are hard to read
- Are there any plots for the main result of section 5.2? I understands all figures and table 1 refer to mTRE, which refers to "no disadvantages for key point registration" with your method. But what about "many advantages with your method to keep cellular structures"? I don't see results here.
- The paper would greatly benefit from an example image after registration, which shows how your method preserves morphological structure, while other methods destroy that (qualitative results).

Minor remarks:
- Figure 1: Please explain more. What is red, what is green? Which original images have been registered? Left to right? To me, the middle green point does not seem to belong the same anatomic region?!
- Page 2: "Fischerand"
- Page 2: "For images with are similar"
- Make all et al. italic
- Page 3: "Like some previous authors, segmentation plays a key role in our registration pipeline." This sentence reads strange.
- Section 3.2: "we use that of Starting et. al Staring et al. (2006)"
- Please review the references. E.g. "Fischerand and", S. S. SHAPRIO (capital letters), make them consistently either full name, or just family name.
- Section 5.1 What does it mean for 10^-2? Are the key points significantly shifted between the two methods?

---

### Official Review · Reviewer_gbUk · 2021-08-20
**Relevant application, but unclear methodology and poor presentation of the results**

**Rating:** 4
**Confidence:** 5

**Review:**

The authors present a method for registration of histopathology slides that is "content-aware", i.e. it uses a regularisation term that aims to prevent non-realistic transformations important structures such as cells. This is a very relevant problem as, indeed, the morphological properties of such structures should be preserved in the registration process in order to properly extract tissue biomarkers.

While the addressed problem is important, I find that the main ideas and results of the paper are not very clearly presented. My major remarks are the following:

- Figure 1 illustrates the main motivation behind the work - preserving cell structures. However, at this magnification level, it is not possible t see the "shape and size" of the cells in the image, only of larger tissue structures (glands?, impossible to tell because proper dataset description is missing, only a reference is provided). Perhaps the authors referred to these larger structures, but in that case they should not be referred as cells.
- The authors should include more figures that illustrate the different components of the models, particularly the cell segmentations and the resulting cellularity maps. Qualitative results are also missing. Including such figures will greatly help with understanding the presented methodology and the achieved result.
- After reading section 3.4, it is precisely clear to me what the regularisation term is. The authors should explicitly define R(\theta).

Some more minor issues:

- SSD and mTRE are not defined. While it is clear from the context that they refer to the sum of square differences and median target registration error, this needs to be explicitly defined
- "kronecker delta, "jacobian" -> "Kronecker delta", "Jacobian"
- The figures, particularly in the results section, should have more descriptive captions
- "set of (no-longer binary) `mask' images" sounds very odd, I suggest "set of soft mask images"
- The method seems to rely on point-based pre-registration. Are these keypoints selected manually? If yes, then this should be emphasized as the method is no longer automatic and has more limited use

---

### Decision · Program_Chairs · 2021-08-25

Reject